# Downregulation of cytoplasmic DNases is implicated in cytoplasmic DNA accumulation and SASP in senescent cells

Akiko Takahashi[1,2], Tze Mun Loo[1,3], Ryo Okada[1], Fumitaka Kamachi[3], Yoshihiro Watanabe[3], Masahiro Wakita[4], Sugiko Watanabe [4], Shimpei Kawamoto[4], Kenichi Miyata[1], Glen N. Barber[5], Naoko Ohtani[3,6] & Eiji Hara[1,4]

Accumulating evidence indicates that the senescence-associated secretory phenotype (SASP) contributes to many aspects of physiology and disease. Thus, controlling the SASP will have tremendous impacts on our health. However, our understanding of SASP regulation is far from complete. Here, we show that cytoplasmic accumulation of nuclear DNA plays key roles in the onset of SASP. Although both DNase2 and TREX1 rapidly remove the cytoplasmic DNA fragments emanating from the nucleus in pre-senescent cells, the expression of these DNases is downregulated in senescent cells, resulting in the cytoplasmic accumulation of nuclear DNA. This causes the aberrant activation of cGAS-STING cytoplasmic DNA sensors, provoking SASP through induction of interferon-β. Notably, the blockage of this pathway prevents SASP in senescent hepatic stellate cells, accompanied by a decline of obesity-associated hepatocellular carcinoma development in mice. These findings provide valuable new insights into the roles and mechanisms of SASP and possibilities for their control.

---

[1] The Cancer Institute, Japanese Foundation for Cancer Research, Koto-ku, Tokyo 135-8550, Japan. [2] PRESTO, JST, Kawaguchi, Saitama 332-0012, Japan. [3] Faculty of Science & Technology, Tokyo University of Science, Noda-shi, Chiba 278-8510, Japan. [4] Research Institute for Microbial Diseases, Osaka University, Suita-shi, Osaka 565-0871, Japan. [5] Department of Medicine, University of Miami Miller School of Medicine, Miami, FL 33136, USA. [6] Graduate School of Medicine, Osaka City University, Abeno-ku, Osaka 545-8585, Japan. These authors contributed equally: Akiko Takahashi, Tze Mun Loo, Ryo Okada. Correspondence and requests for materials should be addressed to E.H. (email: ehara@biken.osaka-u.ac.jp)

Although the irreversible cell-cycle arrest is traditionally believed as the major function of senescent cells[1–5], recent studies have unveiled some additional functions of senescent cells[1–4]. Most noteworthy among them is the increased secretion of various pro-inflammatory proteins, such as inflammatory cytokines, chemokines, and growth factors, into the surrounding extracellular space[6–8]. This newly recognized senescent phenotype, termed the senescence-associated secretory phenotype (SASP)[8], reportedly contributes to tumor suppression[6,7,9], tissue regeneration[10], embryonic development[9,10], and even tumourigenesis promotion[8,11], depending on the biological context[12–18]. Thus, controlling the induction of SASP could profoundly affect the maintenance of homeostasis and disease control. However, although persistent activation of the DNA damage response (DDR), which is believed to drive the cell senescence program, is known to play key roles in the onset of SASP[19,20], the precise mechanisms underlying this process remain largely unclear.

In eukaryotic cells, the localization of self DNA is restricted to the nucleus and mitochondria, and thereby the self DNA is sequestered from the cytoplasmic DNA sensing machineries, which activate pro-inflammatory cytokine pathways[21–25]. In normal, healthy cells, DNase2 and TREX1 (DNase3), cytoplasmic DNases that target double stranded (ds)DNA and single stranded (ss)DNA for degradation, respectively, prevent the cytoplasmic accumulation of free DNA[26–28]. In senescent cells, however, DNA fragments of nuclear origin reportedly accumulated in the cytoplasm[29,30]. Moreover, it has recently become apparent that DNA damage causes the cytoplasmic accumulation of nuclear DNA in various cell types[28,30,31]. These reports, together with our previous observations that senescent cells express increased levels of interferon (IFN)-β[30,32], a pro-inflammatory cytokine known to be induced by the cytoplasmic DNA sensing pathway[21–25], led us to the idea that persistent DDR activation may provoke SASP through the aberrant activation of the cytoplasmic DNA sensing pathway, at least to a certain extent, in senescent cells.

In the present study, we reveal that although both dsDNA and ssDNA are constitutively emanating from the nucleus to the cytoplasm, DNase2 and TREX1 remove the exported nuclear DNA rapidly before it accumulates, thereby preventing the aberrant activation of the cytoplasmic DNA sensing pathway and consequently SASP in pre-senescent cells. However, in senescent cells, the downregulation of DNase2 and TREX1 expression appears to cause the cytoplasmic accumulation of nuclear DNA, thus provoking SASP through the aberrant activation of the cGAS-STING cytoplasmic DNA sensing machinery. Interestingly, moreover, the blockage of this pathway prevents SASP in senescent hepatic stellate cells, accompanied by a decline of obesity-associated hepatocellular carcinoma development in mice. These results strongly suggest that the down-regulation of DNase2/TREX1 is contributing to the activation of the cGAS/STING pathway and the consequent induction of SASP, at least to a certain extent in senescent cells in vivo.

## Results

### Activation of cytoplasmic DNA sensing pathway causes SASP.
To explore the idea that persistent DDR activation may provoke SASP through the aberrant activation of the cytoplasmic DNA sensing pathway, we first assessed whether the cytoplasmic DNA sensing machineries are activated in senescent cells. Pre-senescent (early-passage) normal human diploid fibroblasts (HDFs) were rendered senescent by either serial passage or ectopic expression of oncogenic Ras, the most established ways to induce cellular senescence in cultured cells[1–4], and then we examined the levels of the phosphorylated (activated) forms of TBK1 and IRF3,

downstream mediators of the cGAS-STING cytoplasmic DNA-sensing pathway[21,33–35]. Indeed, the levels of phosphorylated TBK1 and IRF3 were substantially increased, although the levels of cGAS and STING were not substantially changed in senescent HDFs, regardless of how the cellular senescence was induced (Fig. 1a–c, and Supplementary Fig. 1). This was accompanied by the cytoplasmic accumulation of nuclear DNA and the increased expression of IFN-β, a downstream mediator of the cytoplasmic DNA sensing machinery, and the other SASP factors (Fig. 1d, e and Supplementary Fig. 2). Notably, moreover, siRNA-mediated depletion of STING or cGAS, using previously validated siRNA oligos[30,35], resulted in a substantial reduction in the levels of the phosphorylated forms of TBK1 and IRF3, accompanied by a striking decline of SASP factor expression in Ras-induced senescent HDFs (Fig. 2a, b). Note that although the signs of DDR, such as DNA damage foci, intracellular levels of ROS, phosphorylation of p53 and the levels of p21$^{Waf1/Cip1}$ and p16$^{INK4a}$ expression, were decreased to some extent by depletion of cGAS or STING in senescent cells, cell-cycle was still arrested in these experimental condition, precluding the possibility that the reduction of SASP factor expression was simply due to abolishment of senescence cell-cycle arrest (Fig. 2). These results together with the observations that the treatment of pre-senescent HDFs with IFN-β provoked phosphorylation (activation) of NFκB, a transcription factor required for the expression of numerous SASP factors[6] and induction of other SASP factors expression (Supplementary Fig. 3), suggest that the aberrant activation of the cGAS-STING cytoplasmic DNA sensing machinery and the subsequent activation of the IFN-β pathway play key role(s) in SASP induction, at least to some extent in cultured primary HDFs.

### Repression of DNase2/TREX1 causes cytoplasmic DNA accumulation.
These notions then raised the question of how nuclear DNA accumulates in the cytoplasm of senescent cells. In seeking an explanation, we noted that the expression levels of both DNase2 and TREX1 were markedly reduced in senescent HDFs (Fig. 1a). Moreover, this also occurred when cellular senescence was induced in other types of normal human cells, such as primary human retinal pigment epithelial (HRPE) and primary human embryonic keratinocyte (HEK) cells, accompanied by the cytoplasmic accumulation of nuclear DNA, the activation of cytoplasmic DNA sensing pathways and the induction of SASP (Supplementary Fig. 4). These results led us to consider that the down-regulation of DNase2 and TREX1 expression may cause the cytoplasmic accumulation of nuclear DNA, resulting in the aberrant activation of the cytoplasmic DNA sensing pathway, which in turn provokes the onset of SASP, at least in certain cultured cell types. To verify this idea, we attempted to reduce the levels of cytoplasmic nuclear DNA by overexpressing DNase2 or TREX1 in senescent HDFs. Indeed, the ectopic expression of DNase2 or TREX1 substantially reduced the levels of cytoplasmic nuclear DNA and alleviated the activation of the cytoplasmic DNA sensing pathway in Ras-induced senescent HDFs (Fig. 3a, b). Notably, these phenomena were also accompanied by significant reductions in the expression of various SASP factors (Fig. 3c), although the intracellular levels of ROS, signs of the DDR, and cell cycle arrest were slightly but not significantly attenuated (Fig. 3d-f). Conversely, the depletion of DNase2 and/or TREX1, using previously validated siRNA oligos[36,37], resulted in the cytoplasmic accumulation of nuclear DNA, followed by the activation of DNA sensing pathways and the subsequent induction of the IFN-β pathway in pre-senescent HDFs (Fig. 4). Interestingly, the simultaneous depletion of DNase2 and TREX1

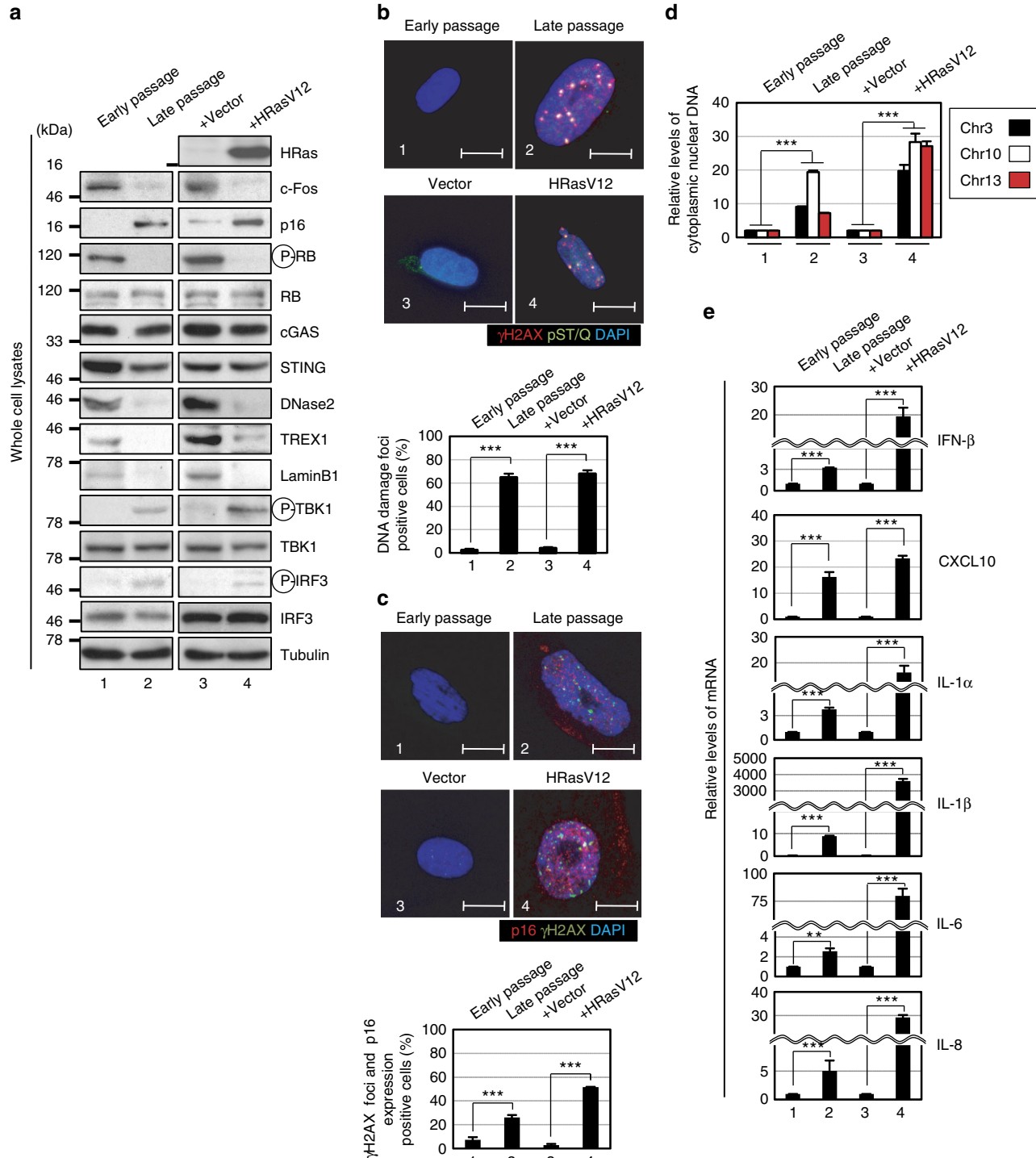

**Fig. 1** Activation of cytoplasmic DNA sensing machineries in senescent cells. **a–e** Pre-senescent TIG-3 cells were rendered senescent by either serial passage (late passage) or ectopic expression of oncogenic *ras* (+HRasV12). These cells were then subjected to western blotting using antibodies shown right (**a**), immunofluorescence staining for markers of DNA damage (γ-H2AX [red], phosphor-Ser/Thr ATM/ATR (pST/Q) substrate [green] and 40,6-diamidino-2-phenylindole [blue]) (**b**) and p16 expression and γ-H2AX foci formation (p16[red], γ-H2AX [green] and 40,6-diamidino-2-phenylindole [blue]) (**c**), qPCR analysis of chromosomal DNA in cytoplasmic fraction (**d**) or RT-qPCR analysis of SASP factor gene expression (**e**). Tubulin was used as a loading control (**a**). For all graphs, error bars indicate mean ± standard deviation (s.d.) of triplicate measurements. **P < 0.01. ***P < 0.001. Scale bars, 10 μm (**b** and **c**)

had a greater effect than either depletion alone, indicating that the accumulation of both ssDNA and dsDNA in the cytoplasm plays roles in the induction of SASP (Fig. 4). Collectively, these results strongly suggest that although both dsDNA and ssDNA are constitutively emanating from the nucleus to the cytoplasm,

DNase2 and TREX1 remove the exported nuclear DNA rapidly before it accumulates, thereby preventing the aberrant activation of the cytoplasmic DNA sensing pathway and consequently SASP in pre-senescent cells. However, in senescent cells, although another mechanism may also be involved[4,29], the downregulation

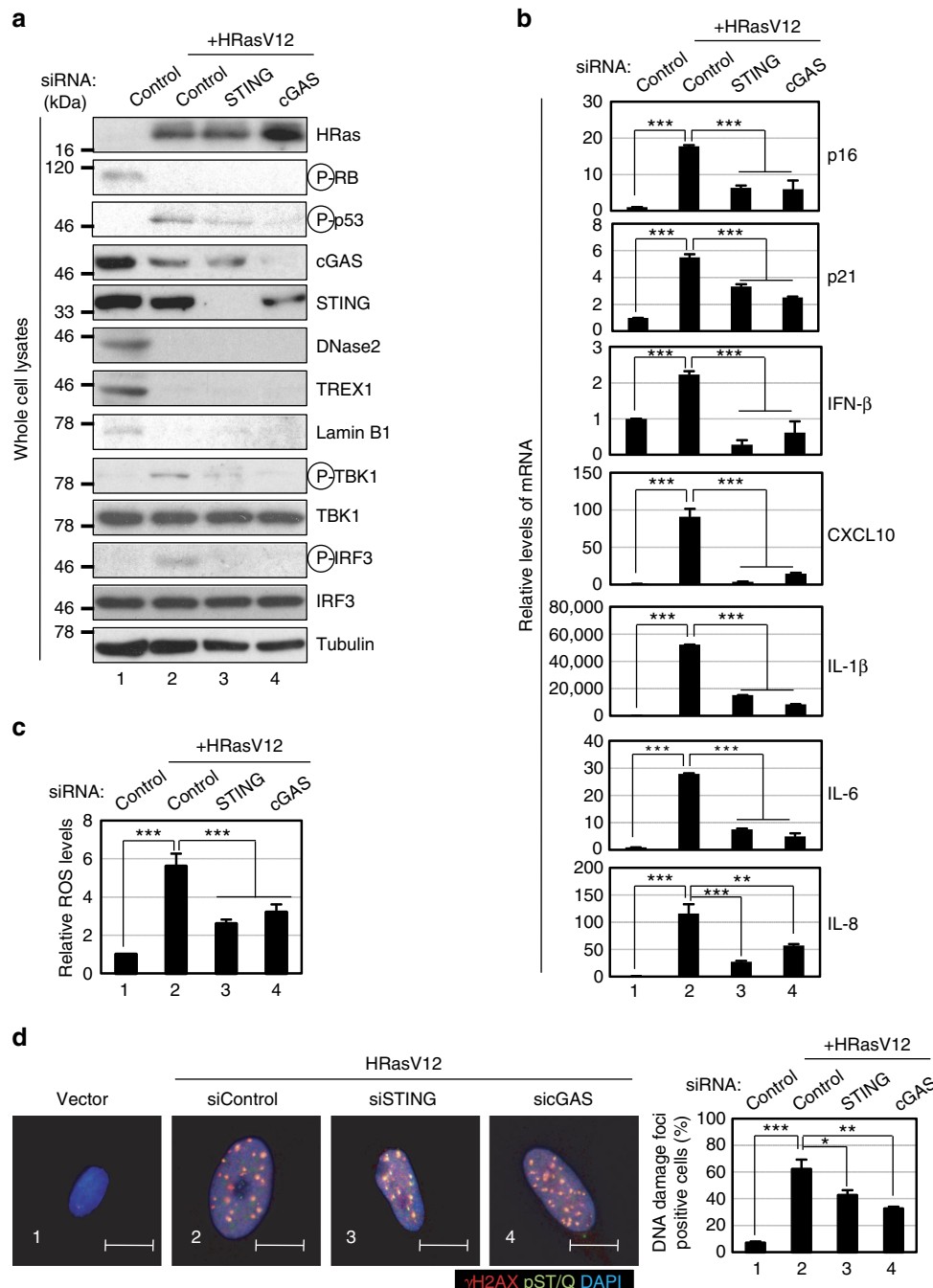

**Fig. 2** Induction of SASP by activating the cytoplasmic DNA sensing pathway. **a** Senescent TIG-3 cells induced by oncogenic Ras expression (lane 2–4) were transfected with previously validated siRNA oligos indicated at the top of the panel for twice at 2 day intervals. These cells were then subjected to western blotting using antibodies shown right (**a**), RT-qPCR analysis of SASP factor gene expression (**b**), analysis of intracellular ROS levels (**c**) or immunofluorescence staining for markers of DNA damage (γ-H2AX [red], phosphor-Ser/Thr ATM/ATR (pST/Q) substrate [green] and 40,6-diamidino-2-phenylindole [blue]) on day 4 (**d**). The representative data from three independent experiments are shown. Tubulin was used as a loading control (**a**). For all graphs, error bars indicate mean ± standard deviation (s.d.) of triplicate measurements. (*$P < 0.05$. **$P < 0.01$. ***$P < 0.001$; one-way ANOVA). Scale bars, 10 μm (**d**)

of DNase2 and TREX1 expression appears to cause the cytoplasmic accumulation of nuclear DNA, thus provoking SASP through the aberrant activation of the cytoplasmic DNA sensing machinery.

**E2F/DP is required for DNase2/TREX1 expression**. To consolidate these ideas, we next explored how the expression of

DNase2 and TREX1 is downregulated during cellular senescence. Interestingly, the levels of the *DNase2* and *TREX1* mRNAs were substantially reduced in senescent cells, regardless of how the cellular senescence was induced in cultured primary cells (Fig. 5a). Moreover, a treatment with MG132 (a proteasome inhibitor) or 3-Methyladenine (an autophagy inhibitor), failed to restore the levels of DNase2 and TREX1 in cultured senescent cells (Fig. 5b and Supplementary Fig. 5), suggesting that the

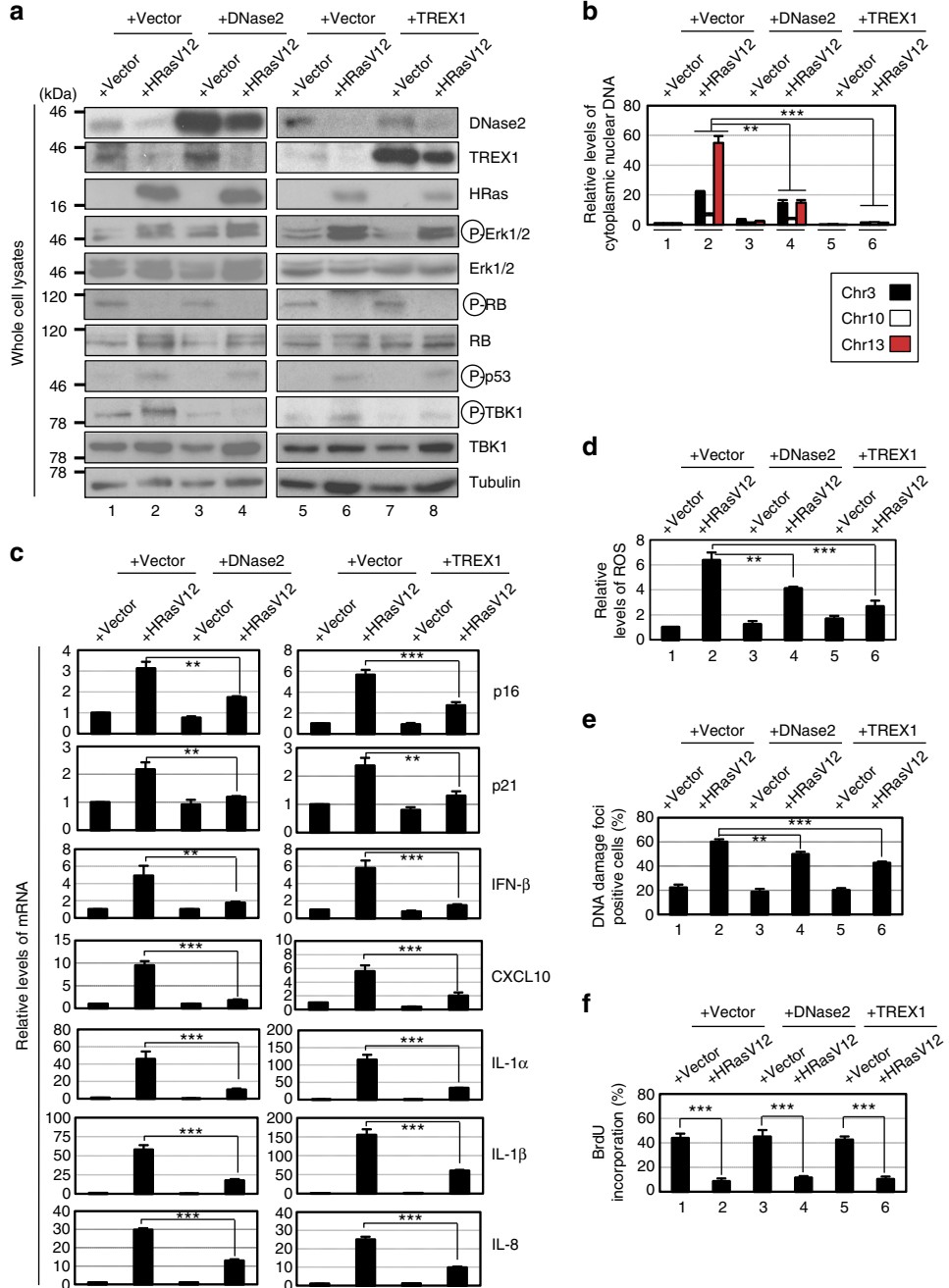

**Fig. 3** Cytoplasmic DNases prevents cytoplasmic accumulation of nuclear DNA and SASP. **a–f** Pre-senescent TIG-3 cells were infected with retrovirus encoding flag-tagged DNase2 (lanes 3 and 4), TREX1 (lane 7 and 8) or empty vector (lanes 1, 2, 5, and 6). After selection with puromycin, cells were super-infected with retrovirus encoding HRasV12 and subjected to western blotting using antibodies shown right (**a**), qPCR analysis for cytoplasmic accumulation of nuclear DNA (**b**), RT-qPCR analysis of SASP factor gene expression (**c**), analysis of intracellular ROS levels (**d**), immunofluorescence staining for markers of DNA damage (γ-H2AX and pST/Q substrate) (**e**) or BrdU incorporation analysis (**f**). Tubulin was used as a loading control (**a**). The histograms indicate the percentage of nuclei that contain more than 3 foci positive for both γ-H2AX and pST/Q staining (**e**). The representative data from three independent experiments are shown. For all graphs, error bars indicate mean ± standard deviation (s.d.) of triplicate measurements. (**P < 0.01. ***P < 0.001; one-way ANOVA)

expression of DNase2 and TREX1 is downregulated at the mRNA level in senescent cells. To obtain mechanistic insight into how the mRNA levels of DNase2 and TREX1 are downregulated during cellular senescence, we next examined the promoter sequences of the *DNase2* and *TREX1* genes. Although the promoter sequences of the *DNase2* and *TREX1* genes are not well conserved between human and mouse, both include potential

binding sites for E2F family transcription factors[38]. Since the transactivation activity of E2F is known to be repressed by the p16[INK4a]-RB pathway in response to persistent DDR[1–5], we tested whether the reduction of E2F activity is responsible for the downregulation of DNase2 and TREX1 expression in senescent cells. Indeed, RNAi-mediated depletion of DP1, an essential activator of the E2F transcription factor, using a previously

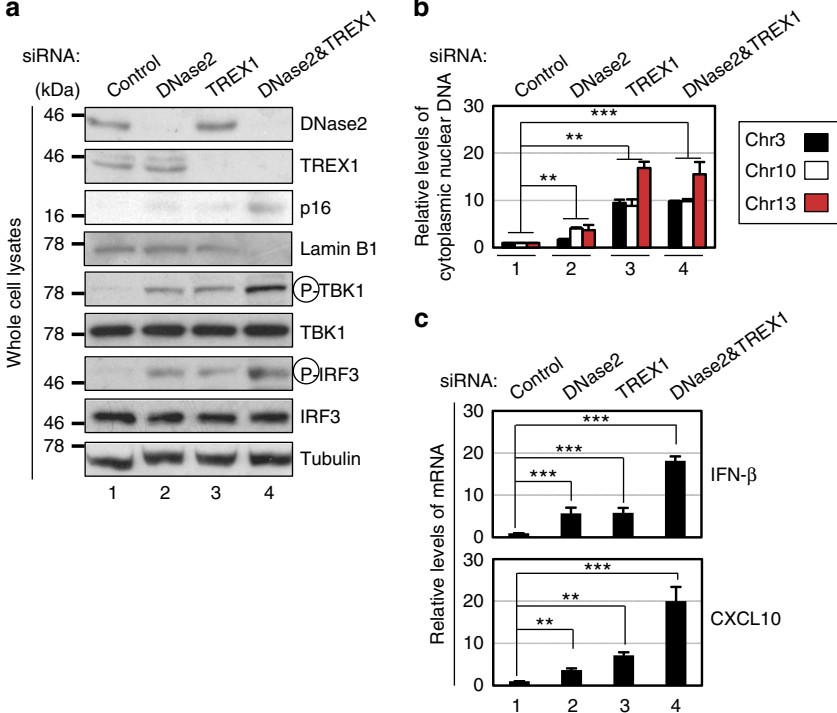

**Fig. 4** The knockdown of cytoplasmic DNases activates the IFN-β pathway. **a–c** Pre-senescent TIG-3 cells were subjected to transfection with indicated siRNA oligos twice (at 2 day intervals). These cells were then subjected to western blotting using antibodies shown right (**a**), isolation of cytoplasmic fraction followed by qPCR analysis of chromosomal DNA (**b**) or qPCR analysis of SASP factor gene expression (**c**). Tubulin was used as a loading control (**a**). The representative data from three independent experiments are shown. For all graphs, error bars indicate mean ± standard deviation (s.d.) of triplicate measurements. (**P < 0.01. ***P < 0.001; one-way ANOVA)

validated shRNA[39] led to significant reductions of *DNase2* and *TREX1* mRNA expression in cultured pre-senescent HDFs (Fig. 5c, d). Furthermore, a chromatin immunoprecipitation (ChIP) analysis revealed that endogenous E2F1 and E2F3, a subset of activator E2Fs, bind to the promoter sequences of the *DNase2* and *TREX1* genes in cultured pre-senescent HDFs (Fig. 5e, f). Together, these results strongly suggest that the expression of DNase2 and TREX1 is maintained, at least partly, through E2F in pre-senescent cells, and the senescence-associated reduction of E2F transcriptional activity results in the cytoplasmic accumulation of nuclear DNA, leading to the aberrant activation of the cytoplasmic DNA sensing pathway, which in turn provokes SASP through the type I IFN pathway in senescent cells.

**cGAS-STING pathway is involved in obesity-associated HCC.** Finally, to extend our findings to in vivo biology, we next sought evidence that the down-regulation of DNase2 and TREX1 expression and the subsequent activation of the cytoplasmic DNA sensing pathway are involved in the onset of SASP in vivo. We previously reported that dietary-obesity or genetic-obesity increases the levels of deoxycholic acid (DCA), a gut bacterial metabolite known to cause DNA damage, in mice[11]. The enterohepatic circulation of DCA provokes SASP in hepatic stellate cells (HSCs), which in turn secrete various inflammatory factors in the liver, resulting in the promotion of hepatocellular carcinoma (HCC) development in obese mice after exposure to a chemical carcinogen. Using this obesity mouse model (Fig. 6a), we found that the expression levels of both DNase2 and TREX1 in HSCs (which express Desmin) were substantially reduced in obese mice fed a high-fat diet (HFD), as compared to those in

lean mice fed a normal diet (ND) (Fig. 6e). This was accompanied by the appearance of senescence markers (53BP1 foci and p21$^{Cip1/Waf1}$ expression), the expression of SASP factors in HSCs and the development of HCC in obese mice fed a HFD (Fig. 6e). However, in obese mice lacking the *sting* gene (*sting*$^{-/-}$ mice)[34], the levels of SASP factor expression in HSCs were substantially reduced and HCC development was also diminished (Fig. 6b, d, e), although the gain of body weight by HFD was similar between control and *sting*$^{-/-}$ mice (Fig. 6c).

Notably, the levels of senescence markers in HSCs were also substantially reduced in obese *sting*$^{-/-}$ mice, as compared to those in obese wild type (WT) mice (Fig. 6e), implying that STING may also be involved in not only the induction of SASP but also the maintenance of cellular senescence by reinforcing DDR in HSCs. These results are somewhat consistent with previous observations that the aberrant activation of the cGAS/STING cytoplasmic DNA sensing pathway promotes reactive oxygen species (ROS) production by activating the IFN-β pathway, thus causing cellular senescence[30,40], and *sting*$^{-/-}$ mice are resistant to DMBA-induced skin carcinogenesis[41]. It should also be noted that the reduction of HCC development is unlikely due to the lack of STING in hepatocytes because STING is not expressed in hepatocytes[42]. Intriguingly, moreover, the expression levels of both DNase2 and TREX1 in HSCs of obese *sting*$^{-/-}$ mice were remarkably increased, as compared to those in obese WT mice (Fig. 6e). This was also the case when cultured primary murine HSCs prepared from *sting*$^{-/-}$ mice fed a ND were rendered senescent by a treatment with DCA and lipoteichoic acid (LTA)[11,43] (Supplementary Fig. 6), accompanied by a striking reduction of the cytoplasmic nuclear DNA

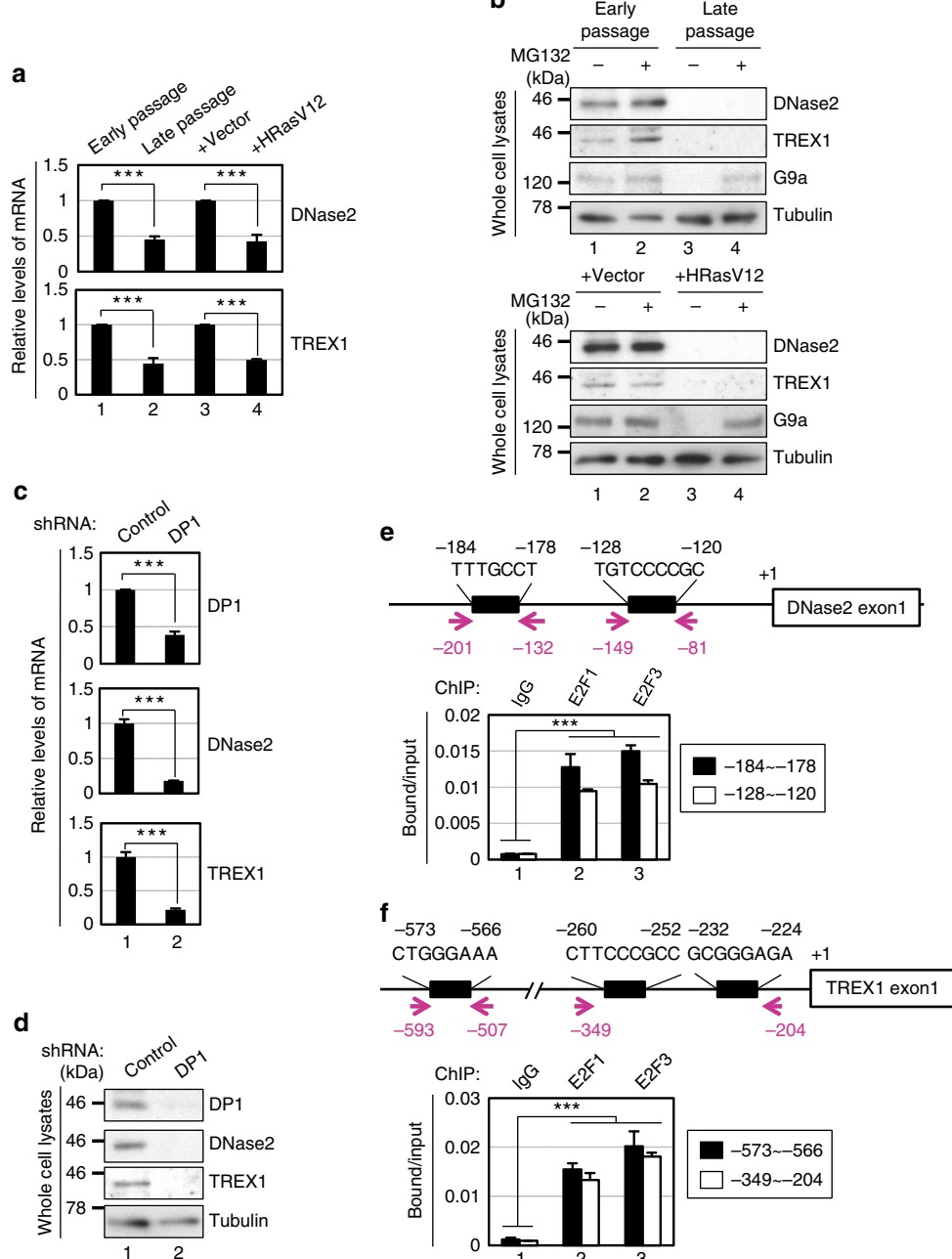

**Fig. 5** E2F-dependent downregulation of DNase2 and TREX1 in senescent cells. **a**, **b** Pre-senescent TIG-3 cells were rendered senescent by either serial passage (late passage) or ectopic expression of oncogenic *ras* (+HRasV12). These cells were then subjected to RT-qPCR analysis of DNase2 and TREX1 gene expressions (**a**), or incubated with 10 µM MG132 for 12 h, then subjected to western blotting using antibodies shown right (**b**). **c**, **d** Pre-senescent TIG-3 cells were infected with retrovirus encoding shRNA against DP1 or control three times. After puromycin selection, these cells were then subjected to RT-qPCR analysis of indicated genes shown right (**c**) or western blotting using antibodies shown right (**d**). Pre-senescent TIG-3 cells were subjected to ChIP analysis using antibodies shown at top and PCR primers of *DNase2* promoter or *TREX1* promoter shown as red arrows in the scheme (**e**, **f**). Tubulin was used as a loading control (**b**, **d**). The representative data from three independent experiments are shown. Error bars indicate mean ± standard deviation (s.d.) of triplicate measurements. (***$P < 0.001$; one-way ANOVA)

(Supplementary Fig. 6b) and the intracellular levels of ROS (Supplementary Fig. 6c). These results imply that a feedback loop may exist between the expression of cytoplasmic DNases and the STING pathway in HSCs. Taken together, these results strongly suggest that the downregulation of DNase2 and TREX1 expression and the consequent activation of the cytoplasmic DNA sensing pathway are likely to play key role(s) in the onset of

both cellular senescence and SASP, at least partly in HSCs under obese conditions in vivo (see model in Fig. 7).

## Discussion
Our findings from the present study fill some gaps between the DDR and SASP induction. Our data indicate that DDR provokes aberrant accumulation of cytoplasmic DNA of nuclear origin at

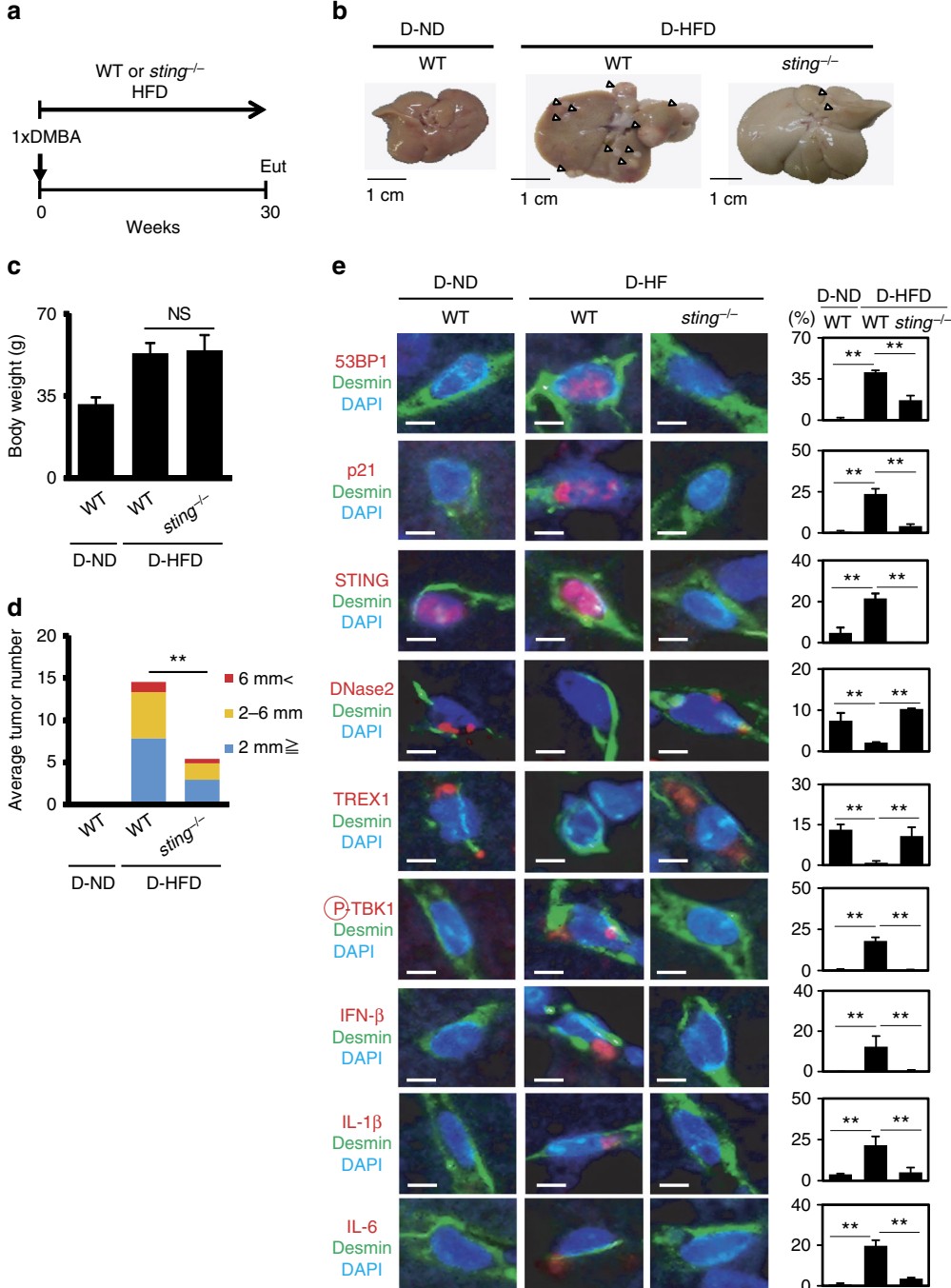

**Fig. 6** Cytoplasmic DNA sensing pathway provokes SASP in HSCs. **a** Timeline of the experimental procedure (ND in WT, $n = 27$; HFD in WT, $n = 24$; HFD in sting$^{-/-}$, $n = 17$). Eut, euthanasia. D, DMBA-treated. **b** Macroscopic photograph of livers of wild-type mice kept with ND (left), with HFD (middle) or sting$^{-/-}$ mice kept with HFD (right) for 30 weeks after the administration of DMBA. The arrowheads indicate HCCs. **c** The average body weight of each group at the age of 30 weeks. **d** The average liver tumor numbers and the relative size distribution (classified as >6, 2–6, ≤2 mm). **e** Immunofluorescence analysis of liver section. HSCs were visualized by Desmin staining (green) and the cell nuclei were stained by 4,6-diamidino-2-phenylindole (DAPI; blue). The histograms indicate the percentages of Desmin expressing cells that were positive for indicated markers. Scale bars, 25 μm. Data of three to four individual mice in each group are represented as means ± SD. More than 100 cells in total were counted for statistical analysis. NS, not significant. (**$P < 0.01$; one-way ANOVA)

least partly through downregulation of DNase2 and TREX1. This event results in the aberrant activation of cGAS-STING cytoplasmic DNA sensors, which in turn provoke SASP through activating the IFN-β signalling pathway (see model in Fig. 7). While our manuscript was being prepared or under review, several papers describing the roles of the cGAS-STING cytoplasmic DNA sensing pathway in cellular senescence and SASP were

published[44–46]. However, it has remained unclear exactly how the cGAS-STING cytoplasmic DNA sensor is activated in senescent cells. Here we show that the downregulation of DNase2/Trex1 plays key roles in the activation of cGAS-STING, at least to some extent, in senescent cells.

However, there might be a debate over whether the downregulation of DNase2 and TREX1 is the cause or the outcome of

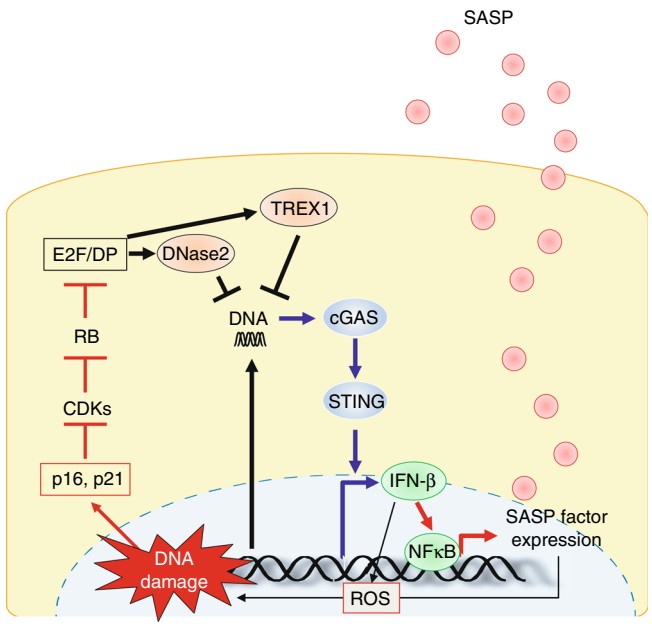

**Fig. 7** A model of SASP induction. The expression of DNase2 and TREX1 is maintained, at least partly, through E2F in pre-senescent cells, and senescence-associated reduction of E2F transcriptional activity results in cytoplasmic accumulation of nuclear DNA, leading to an aberrant activation of cytoplasmic DNA sensing pathway, which in turn provoke SASP through IFN-β pathway in senescent cells

SASP. It has been reported that SASP reinforces senescence cell-cycle arrest[6,7]. For instance, IL-6, a major SASP factor, is known to elevate the intracellular levels of ROS, which are causal factors for DNA damage[47]. Moreover, our group and others have reported that IFN-β provokes DDR through ROS, and thus causes senescence-like cell cycle arrest in HDFs[30,40,48]. Therefore, it is conceivable that once SASP is provoked by DDR, the SASP factors in turn increase the levels of ROS and elicit the positive feedback activation of DDR, thereby amplifying the senescence phenotypes. Taking these points into consideration, it could be possible that the reduction of SASP factor expression, by the overexpression of DNase2/TREX1 or the depletion of cGAS/STING, is due to the abolishment of senescence cell-cycle arrest, but not the direct effects of blocking the cytoplasmic DNA sensing machinery (Figs. 2, 3 and 6e). However, it should be noted that although the signs of DDR, such as DNA damage foci, intracellular levels of ROS, phosphorylation of p53, and the levels of p21$^{Waf1/Cip1}$/p16$^{INK4a}$ expression, were decreased to some extent by the depletion of cGAS/STING or the overexpression of DNase2/TREX1 in senescent cells, the cell-cycle was still arrested under these experimental conditions, precluding the possibility that the reduction of SASP factor expression was simply due to the abolishment of senescence cell-cycle arrest (Figs. 2 and 3). Moreover, the levels of DNase2 and TREX1 were unchanged when cGAS or STING was depleted in Ras-induced senescent HDFs (Fig. 2a), indicating that the down-regulation of DNase2 and TREX1 is not simply an outcome of SASP in senescent cells.

In contrast to our observations, several studies have previously reported that TREX1 is upregulated as the immediate early DDR[49–51] and that such induction of TREX1 occurs via cytokine-induced transcription factors such as STAT1 and AP-1[51,52]. However, it should be noted that cellular senescence occurs in the late state of the DDR and the expression levels of c-FOS, a critical component of AP1, is reportedly reduced in senescent cells[53] (see

also Fig. 1a). Thus, it is quite reasonable that TREX1 expression is reduced, rather than increased, in senescent cells. It is also important to note that unlike HSCs prepared from *sting*$^{-/-}$ mice (Fig. 6e and Supplementary Fig. 6b), HDFs depleted of cGAS or STING by siRNA failed to increase the expression levels of DNase2 and TREX1 (Fig. 2a). We do not know why we did not see any increase of DNase2 and TREX1 expression in this experimental setting. However, unlike the results shown in Fig. 6e and the Supplementary Fig. 6a, cGAS and STING are knocked down, but not knocked out, in Fig. 2a. It is therefore possible that the residual activity of the cGAS and STING pathway may be preventing the upregulation of DNase2 and TREX1 in Fig. 2a. So, the question is whether or not the downregulation of DNase2/TREX1 has some causal effects on the activation of the cGAS/Sting pathway and subsequently the SASP pathway in senescent cells. Importantly, the depletion of DNase2/TREX1 alone can activate the cGAS/STING pathway and SASP in pre-senescent cells (Fig. 4). Conversely, the ectopic expression of DNase2/TREX1 attenuated the activation of the cGAS/STING pathway and SASP in senescent cells (Fig. 3). These results strongly suggest that the down-regulation of DNase2/TREX1 is contributing to the activation of the cGAS/STING pathway and the consequent induction of SASP, at least to a certain extent in senescent HDFs.

However, it is clear that all aspects of SASP induction cannot be explained by the factors described here, and that SASP is subjected to multiple levels of control, such as by signalling molecules (p38, mTOR and Notch)[54–57], transcription factors (NFkB, c/EBP-β, GATA4)[6,7,58] and epigenetic regulators (G9a, GLP, MacroH2A, HMGB2, MLL1, BRD4, H2A.J)[20,59–63], depending on the cellular context. Nevertheless, considering the published data and our findings, it is clear that the DDR-associated downregulation of the expression of the *DNase2* and *TREX1* genes plays key roles in the induction of IFN-β expression and the subsequent activation of NFkB, a critical regulator of SASP[6] in various types of senescent cells, providing new insights into the regulation of SASP (Fig. 7). Thus, in addition to establishing the hierarchy and interplay of these regulatory factors, it will be important to determine whether additional factors are involved in the cytoplasmic accumulation of nuclear DNA in senescent cells.

## Methods

**Cell culture**. TIG-3 cells[5,20,30] were obtained from Japanese Cancer Research Resources Bank (JCRB) and mouse primary HSCs were isolated from mouse liver as previously described[11]. These cells were cultured in Dulbecco's Modified Eagle's (DME) medium supplemented with 10% fetal bovine serum (FBS). HRPE cells (Lonza Inc.) and HEK cells (Cell Applications Inc.) were cultured according to the manufacturer's instructions. Early passage TIG-3 cells (less than 40 population doublings) were used as growing cells, and late passage TIG-3 cells (more than 70 population doublings) that ceased proliferation were used as replicative senescent cells. For retroviral infection, TIG-3 cells were rendered sensitive to infection by ecotropic retroviruses[5]. Cells were then infected with recombinant retroviruses encoding Ras$^{V12}$ or TREX1(in pBabe–puro), DNase2(in pMarX–puro) cDNA or shRNA against *DP1* (in pRetrosuper–puro)[39]. After selecting infected cells by puromycin selection, pools of drug-resistant cells were analyzed 7 days after infection. In some experiments, cells were treated with MG132 (Calbiochem) or recombinant IFN-β (PBL Assay Science) for 12 h. In BrdU incorporation analysis, cells were treated with 20μM BrdU (BD Pharmingen) for 16 h[39]. HSCs were incubated with deoxycholic acid (Wako) for 6 days to induce cellular senescence and then LTA (InvivoGen) for 6 h before harvest[43]. We have confirmed the absence of mycoplasma contamination in our tissue culture cells.

**Isolation of cytosol DNA fractions**. Cytoplasmic DNA was prepared by modifying a method as previously reported[64]. In brief, cells were centrifuged for 1 min in a microcentrifuge, then suspended in 0.3 M sucrose buffer and homogenized with pipetting. The homogenate was overlaid on the same amount of 1.5 M sucrose buffer and centrifuged at 18,506 ×g for 10 min. Cytoplasmic DNA was purified by 0.4 mg/ml Proteinase K (Wako) treatment, phenol/chloroform extraction and ethanol precipitation with a carrier (Dr. GenTLE® Precipitation Carrier, Takara Bio

Inc.). The amount of nuclear DNA was determined by quantitative Real-Time PCR, using three different sets of primers designed for different chromosomes (human chromosome 3, 10, and 13, or mouse chromosome 1, 2, and 5)[30].

**Fluorescence microscopic analysis.** Immunofluorescence analysis was performed using antibodies against Lamin B1 (Abcam, ab16048, 1:1000 dilution), dsDNA marker (Santa Cruz, sc-58749, 1:250 dilution), γ-H2AX (Millipore, 05-636, 1:2000 dilution), phosphor-(Ser/Thr) ATM/ATR substrate (Cell Signaling Technology, 2851, 1:500 dilution), p16 (Santa Cruz, sc-468, 1:500 dilution), p21 (Abcam, ab2961, 1:50 dilution), 53BP1 (Santa Cruz, sc-22760; Abcam, ab36823, 1:500 dilution), IL-1β (Proteintech, 16806-1-AP, 1:300 dilution), IL-6 (Abcam, ab6672, 1:400 dilution), STING (Abcam, ab92605, 1:200 dilution), phospho-TBK1 (Bioss antibodies, bs-3440R, 1:100 dilution), IFN-β (Abcam, ab140211, 1:100 dilution), DNase2 (Bioss antibodies, bs-7652, 1:100 dilution), TREX1 (Novus Biologicals, NBP1-76977, 1:100 dilution) and Desmin (Thermo Fisher Scientific, MA5-13259, 1:100 dilution). DNA was stained with DAPI (Dojindo). Fluorescence images were observed and photographed using an immunofluorescence microscope (Carl Zeiss AG)[11,43].

**RNAi.** RNAi was performed by the transfection of siRNA oligos using the Lipofectamine™ RNAiMAX transfection reagent (Thermo Fisher Scientific), according to the manufacturer's instructions. The sequences of the siRNA oligos were as follows. cGAS: GGAAGGAAAUGGUUUCCAA[35]. DNase2: CAA-GAACCCUGGAACAGCAGCAUCA[36]. TREX1: CCAAGACCAUCUGCU-GUCA[37]. ON-TARGETplus siRNAs to target Sting mRNA sequences and non-targeting control siRNA were used (Dharmacon)[30]. Knockdown efficiency was confirmed by quantitative real-time PCR.

**Plasmids.** The epitope tagged cDNAs of DNase2 was cloned into the pMarX—puro retrovirus vector[5,20,30] or TREX1 was cloned into the pBabe—puro retrovirus vector. To simplify flag-tagged cDNA, the following primers were used: *DNase2*, 5′-ACCGGATCCACCGCCATGGACTACAAAGACGATGACGACAA-GATCCCGCTGCTGCTGGC-3′ (forward) and 5′-ACCCTCGAGTTA-GATCTTATAAGCTCTGCTG-3′ (reverse); and *TREX1*, 5′-ACCGGATCCACCGCCATGGACTACAAAGACGATGACGACAAGCA-GACCCTCATCTTTTTC-3′ (forward) and 5′-ACCGAATTCCTACTCCC-CAGGTGTGGCCAG-3′ (reverse). All cDNAs were sequenced on a Genetic Analyzer 3130 (Applied Biosystems) using a BigDye Terminator v3.1 Cycle Sequencing Kit (Applied Biosystems).

**Quantitative real-time PCR.** Total RNA was extracted from cultured cells using a mirVana kit (Thermo Fisher Scientific), and then subjected to reverse transcription using a PrimeScript RT reagent kit (Takara Bio Inc.). Quantitative real-time RT-PCR was performed on a StepOnePlus PCR system (Applied Biosystems) using SYBR Premix Ex Taq (Takara Bio Inc.)[20,30]. The PCR primer sequences were listed in Supplementary Table 1. The means ± s.d. of three independent experiments are shown.

**Western blotting.** For western blotting analysis, cells or mouse liver tissues were lysed in lysis buffer (50 mM Hepes, pH 7.5, 150 mM NaCl, 1 mM EDTA, 2.5 mM EGTA, 10% glycerol, 0.1% Tween20, 10 mM β-glycerophosphate) with 1% Protease inhibitor cocktail (Nacalai Tesque)[5,20,30]. The protein concentration was determined using a DC Protein Assay (Bio-Rad), and proteins were separated by SDS-PAGE and transferred onto a PVDF membrane (EMD Millipore). After blocking with 5% milk, membranes were probed with primary antibodies as follows: anti-H-Ras (Santa Cruz, sc-29, 1:500 dilution), c-Fos (Santa Cruz, sc-52, 1:500 dilution), p16 (IBL, 11104, 1:250 dilution), phospho-RB (Cell Signaling Technology, 9308, 1:500 dilution), RB (Santa Cruz, sc-102, 1:500 dilution), STING (Cell Signaling Technology, 13647, 1:250 dilution), cGAS (Cell Signaling Technology, 15102, 1:250 dilution), DNase2 (ProSci, 2059, 1:500 dilution), TREX1 (Cell Signaling Technology, 12215, 1:250 dilution), Lamin B1 (Santa Cruz, sc-6217, 1:500 dilution), phospho-TBK1 (Ser172) (Cell Signaling Technology, 5483, 1:250 dilution), TBK1 (Cell Signaling Technology, 3013, 1:250 dilution), phospho-IRF3 (Ser396) (Cell Signaling Technology, 4947, 1:250 dilution), IRF3 (Cell Signaling Technology, 11904, 1:250 dilution), phospho-Erk1/2 (Ser396) (Cell Signaling Technology, 4376, 1:1000 dilution), Erk1/2 (Cell Signaling Technology, 4695, 1:1000 dilution), phospho-NFκB p65 (Ser536) (Cell Signaling Technology, 3031, 1:250 dilution), G9a (Cell Signaling Technology, 3306, 1:200 dilution), NFκB p65 (Santa Cruz, sc-8008, 1:500 dilution), DP1 (ICRF), p62 (MBL, PM045, 1:500 dilution), LC3 (MBL, M152-3, 1:500 dilution), and α-tubulin (SIGMA, T9026, 1:2000 dilution). The membranes were incubated with secondary antibodies (GE Healthcare) and visualized with the SuperSignal West Femto Maximum Sensitivity Substrate (Thermo Fisher Scientific).

**Measurement of intracellular ROS levels.** To assess the levels of intracellular ROS generation, cells were incubated with 20 μM DCF-DA (Calbiochem) at 37 °C for 20 min. The peak excitation wavelength for oxidized DCF (488 nm) and that for emission (525 nm) were measured by using a Wallac ARVO 1420 Multilabel counter (PerkinElmer Co., Ltd.)[5,65].

**Chromatin immunoprecipitation (ChIP) analysis.** ChIP analysis was performed using Dynabeads® Protein G (Thermo Fisher Scientific, 10004D) according to the manufacturer's instruction. Briefly, chromatin was extracted from TIG-3 cells, cross-linked with formaldehyde (final concentration; 1%), and sonicated (Bioruptor, Cosmo Bio Co. Ltd.: 10 cycles of 30 s on/30 s off, on the highest setting) to generate DNA fragments[20]. The immunoprecipitation of cross-linked chromatin was conducted with anti-human E2F1 (Santa Cruz, sc-193X, 1:1000 dilution), anti-human E2F3 (Santa Cruz, sc-878X, 1:1000 dilution) and rabbit IgG (Cell Signaling Technology, 2729, 1:1000 dilution) as a negative control[66]. After immunoprecipitation, DNA was extracted using the QIAquick PCR purification kit (Qiagen) and an aliquot was amplified by real time qPCR using following primers flanking the putative human E2F-binding site position at −184 to −178 bp of human *DNase2* gene promoter: 5′-CAGACTCAGCGTTGCCTTTT-3′ and 5′-CGAGGGTACA-GACTCCTCCC-3′ or primers flanking the putative human E2F-binding site position at −128 to −120 of human *DNase2* gene promoter: 5′-GAG-GAGTCTGTACCCTCGTG-3′ and 5′-TGGCGCCTTTCACTTCCCTA-3′ or primers flanking the putative human E2F-binding site position at −573 to −566 of human *TREX1* gene promoter: 5′-CAGGCCTAAACCAGGAAAGC-3′ and 5′-TCTTACACACTGCTGGGAGC-3′ or primers flanking the putative human E2F-binding site position at −260 to −252 and −232 to −224 of human *TREX1* gene promoter: 5′-ATGGTGGTGAGAGGGACAGAC-3′ and 5′-AGTAGT-GACTCGGCTGCACA-3′[66].

**Animal experiments.** C57/BL6 mice were purchased from CLEA Japan Inc. The *sting*^(−/−) mice (C57/BL6) were kindly provided by Dr. Keiyo Takubo (National Center for Global Health and Medicine)[34]. 7,12-dimethylbenz [a]anthracene (SIGMA) treatments were performed as described previously[11,43]. In brief, 50 μl of a solution of 0.5% DMBA in acetone was applied to the dorsal surface of mice on postnatal days 4, 5. After this application, mother mice with pups were fed ND (NOSAN, Labo MR) or high-fat diet (Research Diet Inc., D12492). At the age of 4 weeks old, pups were weaned and continuously fed either ND or HFD until euthanized. Male mice with more than 45 g weight at the age of 30 weeks old were used as obese mice for all the experiments. Evaluation of tumor number and size was determined by counting the number of visible tumors and measuring the size of the tumor. The sample size used in this study was determined based on the expense of data collection, and the requirement for sufficient statistical significance. Randomisation and blinding were not used in this study. All animal care was performed according to the protocols approved by the Committee for the Use and Care of Experimental Animals of the Japanese Foundation for Cancer Research.

**Statistical analysis.** Statistical significance was determined using a Student's *t*-test and one-way ANOVA. *P*-values less than 0.05 were considered significant.

**Data availability.** The data that support the findings of this study are available from the corresponding author upon reasonable request.

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

## Acknowledgements

We thank the members of the Hara laboratory for helpful discussions during the preparation of this manuscript. This work was supported in part by grants from Japan Agency of Medical Research and Development (AMED) under grant number JP17gm5010001, 17gm0610002h0406, 17cm0106401h0002 and 17fk0210206h0002, Japan Science and Technology Agency (JST)-PRESTO under grant number JP1005462, Japan Society for the Promotion of Science (JSPS) under grant number 26250028, 25290046, 16H04700, 17K19618, 15K06942, and 15H01462, Uehara Memorial Foundation, Cell Science Research Foundation, Princess Takamatsu Cancer Research Fund, the Vehicle Racing Commemorative Foundation, The Naito Foundation, Takeda Science Foundation, and the Research Foundation for Microbial Diseases of Osaka University.

## Author contributions

E.H. designed the experiments. A.T., T.M.L., R.O., F.K., Y.W., M.W., K.M., and S.W. performed the experiments. S.K., G.N.B., N.O., and E.H. analysed the data. E.H. wrote the manuscript and oversaw the projects.

## Additional information

**Competing interests:** The authors declare no competing interests.

