## [Peer Review File(PDF 551 kb) · Nature Communications]

Reviewers' comments:

Reviewer #1 (Remarks to the Author):

I appreciate that the manuscript by Takahashi et al. has improved since the last submission and the authors have satisfactorily addressed almost all of my points. Still, two important concerns remain which are outlined below. If the suggested revisions are fully implemented, I think this study would be suitable for publication in Nature Communications.

- Re-point-2: Indeed there is a context dependent role of the SASP in different models of cancer. The authors have included sufficient details in their results section when discussing their data from the HCC model and also clarified that their conclusions hold true in a context specific manner. I do not have further reservations on this point.

- Re-point-3:

To determine the presence of senescent cells it is a standard to employ multiple makers, such as the mentioned p16 levels, cell cycle indicators etc. as none of these markers alone are solely specific to senescence, including SAbGal activity.

Again, my point aimed to determine the purity of the senescent cell culture, as this is directly imperative for the conclusion of the results throughout the paper. Perhaps co-immunofluorescence staining for two or three markers such as p21-positive, gH2AX-positive, LaminB-negative fraction could be quantified. To my delight, I see that the authors have started to add such quantification in their new Figure 1b. Please expand to co-stainings and other markers as suggested.

However, I maintain my initial suggestion that the authors should still perform the SAbGal activity experiment and leave assessment of shortcomings of the assay to readers of the journal.

- Re-point-4: It is noted that the interplay between DNA damage-SASP-cell cycle arrest is complex and context dependent. Although the authors' arguments are valid, the authors need to exclude one, much more simple, possibility – cell death of senescent cells after depletion of cGAS/STING or the overexpression of DNase2/TREX1. Since a senescent cell culture is heterogeneous per se and the authors' senescent cell cultures are not 100% pure (as addressed in re-point 3 and some indications of this shown in Figure 1b), this could explain an overall reduction in senescence markers, esp. qPCR, WB and ROS data that are quantified in a pool of cells and not on single cell level.

With regard to changes in levels of cGAS and STING at the protein level with induction of senescence, if you consider blots in figure 1a and figure 2a, I would say there is discernable decrease in expression of these proteins. I do agree this does not take away the fact that function/signaling downstream of the cGAS-STING pathway is elevated. Therefore, I have no further reservations with this particular observation.

Reviewer #2 (Remarks to the Author):

I had previously recommended publication of this manuscript in Nature Communications and I remain supportive. I had no major technical concerns and the authors now seem to have responded appropriately to the other reviewers.

Reviewer #3 (Remarks to the Author):

I have looked at the revised manuscript by Takahashi et al. and their response both to my comments and that of the other reviewers. Aside from the concern regarding conceptual advancement, the authors have not addressed the substantive issue raised. The authors' responses for most part do not make sense, at least to me. In fact, from their response one is left with the impression of an attempt to gloss over/skirt around the issues. Moreover the inconsistencies between what is claimed vs what is shown across experiments makes it difficult to vouch for the veracity of their claims. Therefore I am hesitant to recommend it for publication. Because the authors have not addressed most of my main concerns I will reiterate them again here below.

1) The proposed DNA damage – induced downregulation of DNase2 and Trex1 as a possible cause of SASP is interesting. However the data in support of this concept is tenuous. For example
i) Original comment: "It is still not clear whether the observed downregulation of Dnase2 and Trex1 is the cause or an outcome of the SASP. In fact, from the data, it appears that downregulation of DNase2 and Trex1 is a consequence rather than the cause of the SASP. For example, the data in Figure 6e and Sup. Fig 5a, d shows that ablation of STING pathways causes increased expression of DNase2 and Trex1. This means that activation of the cGAS-STING pathway contributes to the downregulation of DNase2 and Trex1 expression and not the other way round. This is contrary to the central claim of this article. Similarly, the data in Figure 4, b shows that increased accumulation of cytoDNA (due to silencing of Dnase2 & Trex1) activates p16INK4a. This again places the p16-RB pathway as a downstream consequence of cytoDNA-mediated activation of cGAS-STING, and not vice versa. Although the authors have now included the expression data of DNase2 and Trex1 in STING and cGAS silenced cells in Figure 2a as I requested, the inclusion of these data do not solve the problem but rather adds on the confusion as it contradicts the data in Figure 6e and Sup. Fig 5a, where ablation of STING pathways causes an elevation in the expression of Dnase2 and Trex1".

Yet in their response they claim "Moreover, the levels of DNase2 and TREX1 were unchanged when cGAS or STING was depleted in Ras-induced senescent HDFs (new Fig. 2a), indicating that the down-regulation of DNase2 and TREX1 is not simply an outcome of SASP". Although membrane exposure in Fig. 2a is suboptimal, when analysed carefully it seems that depletion of cGAS or STING is associated with decrease in the expression of Dnase2 and Trex1. Therefore what is claimed in the text and what the data actually shows are different!

ii) Original comment: "I contrast to their observations, a number studies have previously shown that Trex1 is in fact upregulated in response to DNA damage (e.g. Vanpouille-Box C et al., *Nat Commun.* 2017;8:15618, Tomicic MT et al, *Biochim Biophys Acta.* 2013;1833(8), Christmann M et al., *Nucleic Acids Res.* 2010;38(19)) and that such induction of Trex1 occurs via cytokine-induced transcription factors such as STAT1 and Fos/AP-1 (Christmann M et al. *Nucleic Acids Res.* 2010;38(19), Serra M et al. *J Immunol.* 2011; 186(4)). Interestingly, cytokine-induced senescence has also recently been linked to the activation of p16-RB and STAT1 pathways (Braumüller H et al., *Nature.* 2013, 494(7437)). These published findings do not necessarily discount the proposed downregulation of Trex1 and DNase2 in senescent cells. Rather they illustrate that both DNA damage and cytokine signals can regulate the expression of cytoplasmic DNases. Whether upregulation or downregulation most likely depend on the context. In the present study, the authors analysed the expression of cytoplasmic DNases in serially passaged or oncogene-induced senescent cells and conclude that DNA damage induces the downregulation DNase2 and Trex1. What is the immediate impact of DNA damage (e.g. 1-6 hours after exposure DNA damaging agents) on the expression these DNases? This could have been a good control to include, Therefore, given the experimental conditions of their tests, it is not possible to conclude whether the proposed regulation of DNases is directly due to DNA damage per se or an outcome of the SASP. As aforementioned, the data in Figure 2d (now Fig. 4a) and Figure 4e (now Fig. 6e) are

indicative of the latter. In brief, the central message of the manuscript "DNA damage response provokes SASP by downregulating cytoplasmic DNases in senescent cells" is an overstatement". In response to this comment the authors have provided in the rebuttal an experiment where they show the expression of DNase 2, Trex1 and other activation markers during a 2-10 days course of Ras-induced senescence. This does not address the issue. The expression changes they are showed are most likely due to secondary effects and not direct consequences of DNA damage which is precisely what I requested for: 1-6 hours after exposure of cells to DNA damaging agents such as ionizing irradiation.

Point-by-point responses to the reviewers' comments

We would like to thank all three reviewers for their valuable comments and constructive suggestions. We were pleased to know that both reviewers #1 and #2 have recommended the publication of our manuscript in Nature Communications. Regarding reviewer #3, we think there has been some misunderstanding and we have tried to address the issues that she/he raised.

Reviewer #1 (Remarks to the Author):

I appreciate that the manuscript by Takahashi et al. has improved since the last submission and the authors have satisfactorily addressed almost all of my points. Still, two important concerns remain which are outlined below. If the suggested revisions are fully implemented, I think this study would be suitable for publication in Nature Communications.

- Re-point-2: Indeed there is a context dependent role of the SASP in different models of cancer. The authors have included sufficient details in their results section when discussing their data from the HCC model and also clarified that their conclusions hold true in a context specific manner. I do not have further reservations on this point.

Response-1:

Thank you very much.

- Re-point-3:

To determine the presence of senescent cells it is a standard to employ multiple markers, such as the mentioned p16 levels, cell cycle indicators etc. as none of these markers alone are solely specific to senescence, including SAbGal activity.

Again, my point aimed to determine the purity of the senescent cell culture, as this is directly imperative for the conclusion of the results throughout the paper. Perhaps co-immunofluorescence staining for two or three markers such as p21-positive, gH2AX-positive, LaminB-negative fraction could be quantified. To my delight, I see that the authors have started to add such quantification in their new Figure 1b. Please expand to co-stainings and other markers as suggested.

However, I maintain my initial suggestion that the authors should still perform the SAbGal activity experiment and leave assessment of shortcomings of the assay to readers of the journal.

Response-2:

In line with the reviewer's suggestion, we have included co-immunofluorescence staining data for γ H2AX foci and p16 expression in the new Figure 1c. We have also included SA- β -gal staining data in the new Supplementary Figure 1.

• Re-point-4: It is noted that the interplay between DNA damage-SASP-cell cycle arrest is complex and context dependent. Although the authors' arguments are valid, the authors need to exclude one, much more simple, possibility – cell death of senescent cells after depletion of cGAS/STING or the overexpression of DNase2/TREX1. Since a senescent cell culture is heterogeneous per se and the authors' senescent cell cultures are not 100% pure (as addressed in re-point 3 and some indications of this shown in Figure 1b), this could explain an overall reduction in senescence markers, esp. qPCR, WB and ROS data that are quantified in a pool of cells and not on single cell level.

Response-3:

In accordance with the reviewer's suggestion, we have examined whether cell death is induced in senescent cells after the depletion of cGAS or STING. However, we did not see any substantial increase of cell death by the depletion of cGAS or STING, as judged by the Apoptotic/Necrotic Cells Detection Kit (Takara Bio. # PK-CA707-30017) (see the Figure shown below).

Senescent TIG-3 cells induced by oncogenic Ras expression (lanes 2-4) were transfected twice with the previously validated siRNA oligos indicated at the top of the panel at 2 day intervals. These cells were then tested with the Apoptotic/Necrotic Cells Detection Kit (Takara Bio. # PK-CA707-30017). Lanes 1 and 5 are negative and positive controls treated with (lane 5) or without (lane 1) doxorubicin (Dox).

With regard to changes in levels of cGAS and STING at the protein level with induction of senescence, if you consider blots in figure 1a and figure 2a, I would say there is discernable decrease in expression of these proteins. I do agree this does not take away the fact that function/signaling downstream of the cGAS-STING pathway is elevated. Therefore, I have no further reservations with this particular observation.

Response-4:

We thank this reviewer for his/her fair assessment.

Reviewer #2 (Remarks to the Author):

I had previously recommended publication of this manuscript in Nature Communications and I remain supportive. I had no major technical concerns and the authors now seem to have responded appropriately to the other reviewers.

Response-1:

Thank you very much.

Reviewer #3 (Remarks to the Author):

I have looked at the revised manuscript by Takahashi et al. and their response both to my comments and that of the other reviewers. Aside from the concern regarding conceptual advancement, the authors have not addressed the substantive issue raised. The authors' responses for most part do not make sense, at least to me. In fact, from their response one is left with the impression of an attempt to gloss over/skirt around the issues. Moreover the inconsistencies between what is claimed vs what is shown across experiments makes it difficult to vouch for the veracity of their claims. Therefore I am hesitant to recommend it for publication.

Because the authors have not addressed most of my main concerns I will reiterate them again here below.

1) The proposed DNA damage – induced downregulation of DNase2 and Trex1 as a possible cause of SASP is interesting. However the data in support of this concept is tenuous. For example

i) Original comment: “It is still not clear whether the observed downregulation of Dnase2 and Trex1 is the cause or an outcome of the SASP. In fact, from the data, it appears that downregulation of DNase2 and Trex1 is a consequence rather than the cause of the SASP.

For example, the data in Figure 6e and Sup. Fig 5a, d shows that ablation of STING pathways causes increased expression of DNase2 and Trex1. This means that activation of the cGAS-

STING pathway contributes to the downregulation of DNase2 and Trex1 expression and not the other way round.

This is contrary to the central claim of this article. Similarly, the data in Figure 4, b shows that increased accumulation of cytoDNA (due to silencing of Dnase2 & Trex1) activates p16INK4a. This again places the p16-RB pathway as a downstream consequence of cytoDNA-mediated activation of cGAS-STING, and not vice versa.

Response-1:

We would like to stress that cellular senescence occurs in the late state of the DNA damage response. It takes a while to establish the senescent state, and once fully established, the senescent state is maintained by positive feedback pathways such as SASP (Acosta et al., *Cell* 133, 1006-1018 (2008); Kuilman et al., *Cell* 133, 1019-1031 (2008); Wassmann et al., *Circ. Res.* 94, 534-541 (2004); Takahashi et al., *Nat. Cell Biol.* 8, 1291-1297 (2006)). Therefore, it is possible that the p16-RB pathway could be somewhat attenuated if the onset of SASP is blocked, depending on the cellular context. Indeed, other groups (Gluck et al., *NCB* 19, 1061-1070 (2017); Yang et al., *PNAS* 114; E4612-E4620 (2017)) have also reported that the expression levels of p16 and other senescence markers are somewhat reduced when the cGAS/STING pathway is blocked.

So, the question is whether or not the downregulation of DNase2/Trex1 has some causal effects on the activation of the cGAS/Sting pathway and subsequently the SASP pathway in senescent cells. Importantly, the depletion of DNase2/Trex1 alone can activate the cGAS/STING pathway and SASP in pre-senescent cells (Fig. 4). Conversely, the ectopic expression of DNase2/Trex1 attenuated the activation of the cGAS/STING pathway and SASP in senescent cells (Fig. 3). These results strongly suggest that the down-regulation of DNase2/Trex1 is contributing to the activation of the cGAS/Sting pathway and the consequent induction of SASP, at least to a certain extent in senescent HDFs.

Although the authors have now included the expression data of DNase2 and Trex1 in STING and cGAS silenced cells in Figure 2a as I requested, the inclusion of these data do not solve the problem but rather adds on the confusion as it contradicts the data in Figure 6e and Sup. Fig 5a, where ablation of STING pathways causes an elevation in the expression of Dnase2 and Trex1". Yet in their response they claim "Moreover, the levels of DNase2 and TREX1 were unchanged when cGAS or STING was depleted in Ras-induced senescent HDFs (new Fig. 2a), indicating that the down-regulation of DNase2 and TREX1 is not simply an outcome of SASP". Although membrane exposure in Fig. 2a is suboptimal, when analysed carefully it seems that depletion of cGAS or STING is associated with decrease in the expression of Dnase2 and Trex1. Therefore what is claimed in the text and what the data actually shows are different!

Response-2:

We have repeated the western blotting of DNase2/Trex1 in Figure 2a, and found no substantial differences between lane 2 (control) and lane 3 (Sting-knockdown) or lane 4 (cGAS knockdown) (see new Figure 2a). We do not know why we did not see any

increase of DNase2/Trex1 in this experimental setting. However, unlike the results shown in Figure 6e and the new Supplementary Figure 6a, cGAS/STING are knocked down, but not knocked out, in Figure 2a. It is therefore most likely that the residual activity of the cGAS/STING pathway may be preventing the upregulation of DNase2/Trex1 in Figure 2a. We have discussed this point in the revised text on page 15, lines 5 to 12.

ii) Original comment: “In contrast to their observations, a number of studies have previously shown that Trex1 is in fact upregulated in response to DNA damage (e.g. Vanpouille-Box C et al., *Nat Commun.* 2017;8:15618, Tomicic MT et al, *Biochim Biophys Acta.* 2013;1833(8), Christmann M et al., *Nucleic Acids Res.* 2010;38(19)) and that such induction of Trex1 occurs via cytokine-induced transcription factors such as STAT1 and Fos/AP-1 (Christmann M et al. *Nucleic Acids Res.* 2010;38(19), Serra M et al. *J Immunol.* 2011; 186(4)). Interestingly, cytokine-induced senescence has also recently been linked to the activation of p16-RB and STAT1 pathways (Braumüller H et al., *Nature.* 2013, 494(7437)). These published findings do not necessarily discount the proposed downregulation of Trex1 and DNase2 in senescent cells. Rather they illustrate that both DNA damage and cytokine signals can regulate the expression of cytoplasmic DNases. Whether upregulation or downregulation most likely depend on

the context. In the present study, the authors analysed the expression of cytoplasmic DNases in serially passaged or oncogene-induced senescent cells and conclude that DNA damage induces the downregulation of DNase2 and Trex1. What is the immediate impact of DNA damage (e.g. 1-6 hours after exposure to DNA damaging agents) on the expression of these DNases? This could have been a good control to include. Therefore, given the experimental conditions of their tests, it is not possible to conclude whether the proposed regulation of DNases is directly due to DNA damage per se or an outcome of the SASP. As aforementioned, the data in Figure 2d (now Fig. 4a) and Figure 4e (now Fig. 6e) are indicative of the latter. In brief, the central message of the manuscript “DNA damage response provokes SASP by downregulating cytoplasmic DNases in senescent cells” is an overstatement”.

In response to this comment the authors have provided in the rebuttal an experiment where they show the expression of DNase 2, Trex1 and other activation markers during a 2-10 day course of Ras-induced senescence. This does not address the issue. The expression changes they are showing are most likely due to secondary effects and not direct consequences of DNA damage which is precisely what I requested for: 1-6 hours after exposure of cells to DNA damaging agents such as ionizing irradiation.

Response-3:

In accordance with the reviewer’s suggestion, we have examined the levels of DNase2/Trex1 after exposing the cells to DNA damaging agents for 1-6 hours. As shown in the Figure below, we observed a slight increase of Trex1 expression at 1 hour

after X-ray irradiation. This is somewhat consistent with previous reports (Vanpouille-Box C et al., *Nat. Commun.* 2017;8:15618, Tomicic MT et al., *Biochim. Biophys. Acta* 2013;1833(8), Christmann M et al., *Nucleic Acids Res.* 2010;38(19)).

However, it should be noted that cellular senescence is not a state of the immediate early DNA damage response. It takes at least 6 days to detect cellular senescence after oncogenic Ras-expression, as judged by p16 expression in cultured HDFs. Moreover, although Trex1 is reportedly increased by DNA damage via AP1, the expression level of c-fos, a critical component of AP1, is reportedly reduced in senescent cells (Sesadori et al., *Science* 247, 205-209 (1990); also see new Figure 1a). Thus, it is quite reasonable that Trex1 expression is reduced, rather than increased, in senescent cells. To avoid any confusion, we have discussed this point in the revised text on page 14, line 17 to page 15, line 5.

Early passage TIG-3 cells were exposed to 10G X-rays, and the cells were subjected to western blotting using the antibodies shown on the right and to RT-qPCR analyses for the expression of the indicated genes.

REVIEWERS' COMMENTS:

Reviewer #1 (Remarks to the Author):

The current revision in the manuscript by Takahashi et al. addresses partly addressed the concerns that were raised. Two points in particular still need to be satisfactorily addressed.

- Re-point-3:

The co-stain for p16/γ-H2AX in Figure 1c and the SA-βGal staining in Supplemental figure 1 allows readers to interpret the presented results with regard to purity of the senescent cell population. Although the % of cells with markers are somewhat variable, the consensus is clear that the senescent cell culture is not pure (maximum ~85% achieved with SA-beta-Gal stain and minimally ~30% p16+ γ-H2AX+ cells). Considering that only roughly 30% of cells in the senescent cell culture are positive for cytoplasmic DNA, an important, outstanding point to show that cytoplasmic DNA positive cells are indeed the senescent cells in the culture and not the non-senescent cells. I would suggest to perform a similar co-staining as in Suppl. Figure 2 using dsDNA and γ-H2AX or p16 or SA-beta-Gal.

- Re-point-4:

It was recommended that the authors assay cell death of senescent cells after depletion of cGAS/STING or the overexpression of DNase2/TREX1. The authors chose to assess the effect on cell death after cGAS/STING knockdown in oncogene induced senescent cells. The time after knockdown when the cells were collected for analysis is not specified. Additionally, if this data is added as supplemental material, adding this detail to the methods section will be needed.

Reviewer #3 (Remarks to the Author):

I have again looked at the revised manuscript by Takahashi et al. and their responses. Aside from the concern regarding conceptual advancement, the authors have not addressed the substantive issue raised. As I pointed out previously, when read with a cursory eye, this manuscript is almost convincing. However a careful scrutiny reveals glaring inconsistencies, making it difficult to vouch for the veracity of their claims. As explained below, these inconsistencies also raise questions whether the experiment they show have indeed been conducted reproducibly. Therefore more than before, I am very hesitant to recommend it for publication. According to my assessment, the authors have both miss interpreted and overstated their results.

Please see my previous and present comments.

1) Previous Comment: The proposed DNA damage – induced downregulation of DNase2 and Trex1 as a possible cause of SASP is interesting. However the data in support of this concept is tenuous. For example, "...It is still not clear whether the observed downregulation of Dnase2 and Trex1 is the cause or an outcome of the SASP. In fact, from the data, it appears that downregulation of DNase2 and Trex1 is a consequence rather than the cause of the SASP. For example, the data in Figure 6e and Sup. Fig 5a, d shows that ablation of STING pathways causes increased expression of DNase2 and Trex1. This means that activation of the cGAS-STING pathway contributes to the downregulation of DNase2 and Trex1 expression and not the other way round. This is contrary to the central claim of this article. Similarly, the data in Figure 4, b shows that increased accumulation of cytoDNA (due to silencing of Dnase2 & Trex1) activates p16INK4a. This again places the p16-RB pathway as a downstream consequence of cytoDNA-mediated activation of cGAS-STING, and not vice versa.

Comment: The central claim (as summarized in Figure 7) in this study is that SASP is provoked by downregulation of DNase2 and Trex1 resulting from DNA damage-induced activation of the p16-RB pathway. In my view, their data overwhelmingly support the reverse model: downregulation of downregulation of DNase2 and Trex1 is a consequence and the cause of the SASP. Although the

authors have now discussed around this issue, overall, I still find the way they have presented their data is miss-leading.

Previous Comment: Although the authors have now included the expression data of DNase2 and Trex1 in STING and cGAS silenced cells in Figure 2a as I requested, the inclusion of these data do not solve the problem but rather adds on the confusion as it contradicts the data in Figure 6e and Sup. Fig 5a, where ablation of STING pathways causes an elevation in the expression of Dnase2 and Trex1. Yet in their response they claim "Moreover, the levels of DNase2 and TREX1 were unchanged when cGAS or STING was depleted in Ras-induced senescent HDFs (new Fig. 2a), indicating that the down-regulation of DNase2 and TREX1 is not simply an outcome of SASP". Although membrane exposure in Fig. 2a is suboptimal, when analysed carefully it seems that depletion of cGAS or STING is associated with decrease in the expression of Dnase2 and Trex1. Therefore what is claimed in the text and what the data actually shows are different!

Comment:

The previous DNase2 and Trex1 controls in Fig 2a showed that knockdown of STING or cGAS resulted in decreased expression of Dnase2 and Trex1. Yet to buttress their hypothesis the author claimed "Moreover, the levels of DNase2 and TREX1 were unchanged when cGAS or STING was depleted in Ras-induced senescent HDFs (new Fig. 2a), indicating that the down-regulation of DNase2 and TREX1 is not simply an outcome of SASP". This was incorrect, misleading and inconsistent with the data in other experiments (Fig. 6e and the new Supplementary Figure 6a). In response to this concern, the authors now claim they have repeated the experiment. Inexplicably, now they don't see any expression of Dnase2 or Trex1. Yet again they go on to state "HDFs depleted of cGAS or STING by siRNA failed to increase the expression levels of 250 DNase2 and TREX1 (Fig. 2a)". Again, this is misleading since what they are referring to is a comparison of no signal vs no signal. That notwithstanding, as already mentioned these particular data are not consistent with the data in the other experiments (Fig. 6e and the new Supplementary Figure 6a). The authors' response is that they don't know. Then they argue that this could be due to the limitations of shRNA silencing compared to knockout cells. Which of their shRNA experiments should the reader therefore believe? If they themselves have doubts about some components of the data, then why are they included? And by the way, why did they not include the corresponding data in Fig. 2b - these are obvious controls? Back to Fig. 2a: although not optimal, the previous DNase and Trex1 plots in Fig. 2a at least showed some signal. So, on what basis have the authors now decided that these new blots without any signal are the correct ones to include and not the previous ones? This seems like an attempt to cherry pick and present data to detract the reader from the apparent inter-experimental consistencies. This casts serious doubts about the reproducibility of their experiments. They claim that their data are representative of three independent experiments. If that were the case, upon request they should have readily provided an entire new set of data from an independent biological replicate. Instead, they have done the very minimum; present reanalysis of samples from one experiment, which even then turns out to inconclusive.

I can highlight many other examples but I don't believe this is the point of the review process. My part, I believe, is to point out the main issues the authors need to pay attention to and hope they address them fully, and not just the very minimum to go past the reviewer. This does not improve the manuscript and I feel this is the case!

Previous Comment: "In contrast to their observations, a number studies have previously shown that Trex1 is in fact upregulated in response to DNA damage (e.g. Vanpouille-Box C et al., *Nat Commun.* 2017;8:15618, Tomicic MT et al, *Biochim Biophys Acta.* 2013;1833(8), Christmann M et al., *Nucleic Acids Res.* 2010;38(19)) and that such induction of Trex1 occurs via cytokine-induced transcription factors such as STAT1 and Fos/AP-1 (Christmann M et al. *Nucleic Acids Res.* 2010;38(19), Serra M et al. *J Immunol.* 2011; 186(4)). Interestingly, cytokine-induced senescence has also recently been linked to the activation of p16-RB and STAT1 pathways (Braumüller H et al., *Nature.* 2013, 494(7437)). These published findings do not necessarily discount the proposed downregulation of Trex1 and DNase2 in senescent cells. Rather they illustrate that both DNA damage and cytokine signals can regulate the expression of cytoplasmic DNases. Whether

upregulation or downregulation most likely depend on the context. In the present study, the authors analysed the expression of cytoplasmic DNases in serially passaged or oncogene-induced senescent cells and conclude that DNA damage induces the downregulation DNase2 and Trex1. What is the immediate impact of DNA damage (e.g. 1-6 hours after exposure DNA damaging agents) on the expression these DNases? This could have been a good control to include. Therefore, given the experimental conditions of their tests, it is not possible to conclude whether the proposed regulation of DNases is directly due to DNA damage per se or an outcome of the SASP. As aforementioned, the data in Figure 2d (now Fig. 4a) and Figure 4e (now Fig. 6e) are indicative of the latter. In brief, the central message of the manuscript "DNA damage response provokes SASP by downregulating cytoplasmic DNases in senescent cells" is an overstatement". In response to this comment the authors have provided in the rebuttal an experiment where they show the expression of DNase 2, Trex1 and other activation markers during a 2-10 days course of Ras-induced senescence. This does not address the issue. The expression changes they are showed are most likely due to secondary effects and not direct consequences of DNA damage which is precisely what I requested for: 1-6 hours after exposure of cells to DNA damaging agents such as ionizing irradiation.

Comment: I thank the authors for finally performing this control experiment which should be included at least as part of the supplementary figures. Although the authors claim that there is some decrease, clearly there is no decreased expression of Dnase 2 or Trex1 upon DNA damage until after 10 days. This confirms that activation of p16-RB pathway or the downregulation of Dnase 2 and Trex1 is NOT a direct consequence of DNA damage but a secondary outcome of senescence. The authors argue that decrease in Dnase2 and Trex1 is not apparent in cells because "It takes at least 6 days to detect cellular senescence after oncogenic Ras-expression, as judged by p16 expression in cultured HDFs". I take this as an acknowledgement of the same: activation of p16-RB pathway is not directly due to DNA damage but a secondary consequence of senescence.

Point-by-point responses to the reviewers' comments

Reviewer #1 (Remarks to the Author):

The current revision in the manuscript by Takahashi et al. addresses partly addressed the concerns that were raised. Two points in particular still need to be satisfactorily addressed.

- Re-point-3:

The co-stain for p16/y-H2AX in Figure 1c and the SA-βGal staining in Supplemental figure 1 allows readers to interpret the presented results with regard to purity of the senescent cell population. Although the % of cells with markers are somewhat variable, the consensus is clear that the senescent cell culture is not pure (maximum ~85% achieved with SA-beta-Gal stain and minimally ~30% p16+ y-H2AX+ cells). Considering that only roughly 30% of cells in the senescent cell culture are positive for cytoplasmic DNA, an important, outstanding point to show that cytoplasmic DNA positive cells are indeed the senescent cells in the culture and not the non-senescent cells. I would suggest to perform a similar co-staining as in Suppl. Figure 2 using dsDNA and y-H2AX or p16 or SA-beta-Gal.

Response:

We agree that this is an important point, and have therefore included co-immunostaining data for cytoplasmic DNA, γ-H2AX and p16^{INK4a}, confirming that most of the cytoplasmic DNA positive cells are indeed the senescent cells (see new Supplementary Fig. 2). We thank this reviewer for this important suggestion.

- Re-point-4:

It was recommended that the authors assay cell death of senescent cells after depletion of cGAS/STING or the overexpression of DNase2/TREX1.

The authors chose to assess the effect on cell death after cGAS/STING knockdown in oncogene induced senescent cells. The time after knockdown when the cells were collected for analysis is not specified. Additionally, if this data is added as supplemental material, adding this detail to the methods section will be needed.

Response:

We apologize for omitting this information. Senescent TIG-3 cells induced by oncogenic Ras expression were transfected with validated siRNAoligos (si-cGas, si-STING) twice, at 2 day intervals. These cells were then subjected to the apoptosis analysis on day 4. However

since these data are just for the editor and reviewers and are not included in the supplemental materials, we did not add this information in the method section.

Reviewer #3 (Remarks to the Author):

I have again looked at the revised manuscript by Takahashi et al. and their responses. Aside from the concern regarding conceptual advancement, the authors have not addressed the substantive issue raised. As I pointed out previously, when read with a cursory eye, this manuscript is almost convincing. However a careful scrutiny reveals glaring inconsistencies, making it difficult to vouch for the veracity of their claims. As explained below, these inconsistencies also raise questions whether the experiment they show have indeed been conducted reproducibly. Therefore more than before, I am very hesitant to recommend it for publication. According to my assessment, the authors have both miss interpreted and overstated their results.

Please see my previous and present comments.

1) Previous Comment: The proposed DNA damage – induced downregulation of DNase2 and Trex1 as a possible cause of SASP is interesting. However the data in support of this concept is tenuous.

For example, “...It is still not clear whether the observed downregulation of Dnase2 and Trex1 is the cause or an outcome of the SASP. In fact, from the data, it appears that downregulation of DNase2 and Trex1 is a consequence rather than the cause of the SASP. For example, the data in Figure 6e and Sup. Fig 5a, d shows that ablation of STING pathways causes increased expression of DNase2 and Trex1. This means that activation of the cGAS-STING pathway contributes to the downregulation of DNase2 and Trex1 expression and not the other way round. This is contrary to the central claim of this article. Similarly, the data in Figure 4, b shows that increased accumulation of cytoDNA (due to silencing of Dnase2 & Trex1) activates p16INK4a. This again places the p16-RB pathway as a downstream consequence of cytoDNA-mediated activation of cGAS-STING, and not vice versa.

Comment: The central claim (as summarized in Figure 7) in this study is that SASP is provoked by downregulation of DNase2 and Trex1 resulting from DNA damage-induced activation of the p16-RB pathway. In my view, their data overwhelmingly support the reverse model: downregulation of DNase2 and Trex1 is a consequence and the cause of the SASP. Although the authors have now discussed around this issue, overall, I still find the way they have presented their data is miss-leading.

Response:

It should be noted that the depletion of DNase2/Trex1 alone can activate the cGAS/STING

pathway and SASP in pre-senescent cells (Fig. 4). Conversely, the ectopic expression of DNase2/Trex1 attenuated the activation of the cGAS/STING pathway and SASP in senescent cells (Fig. 3). These results strongly suggest that the down-regulation of DNase2/Trex1 is contributing to the activation of the cGAS/Sting pathway and the consequent induction of SASP, at least to a certain extent, in senescent HDFs. Nevertheless, this reviewer did not consider the roles of DNase2/Trex1 in the regulation of the cytoplasmic accumulation of DNA and SASP. We feel like this is kind of a chicken-and-egg problem (which came first: the downregulation of DNase2/3 or the induction of SASP?) that seems to go around in circles. In order to avoid a barren discussion, we have toned down our message by changing the title of our paper, as follows: “Downregulation of cytoplasmic DNases is implicated in cytoplasmic DNA accumulation and SASP in senescent cells”.

Previous Comment: Although the authors have now included the expression data of DNase2 and Trex1 in STING and cGAS silenced cells in Figure 2a as I requested, the inclusion of these data do not solve the problem but rather adds on the confusion as it contradicts the data in Figure 6e and Sup. Fig 5a, where ablation of STING pathways causes an elevation in the expression of Dnase2 and Trex1. Yet in their response they claim “Moreover, the levels of DNase2 and TREX1 were unchanged when cGAS or STING was depleted in Ras-induced senescent HDFs (new Fig. 2a), indicating that the down-regulation of DNase2 and TREX1 is not simply an outcome of SASP”. Although membrane exposure in Fig. 2a is suboptimal, when analysed carefully it seems that depletion of cGAS or STING is associated with decrease in the expression of Dnase2 and Trex1. Therefore what is claimed in the text and what the data actually shows are different!

Comment:

The previous DNase2 and Trex1 controls in Fig 2a showed that knockdown of STING or cGAS resulted in decreased expression of Dnase2 and Trex1. Yet to buttress their hypothesis the author claimed “Moreover, the levels of DNase2 and TREX1 were unchanged when cGAS or STING was depleted in Ras-induced senescent HDFs (new Fig. 2a), indicating that the down-regulation of DNase2 and TREX1 is not simply an outcome of SASP”. This was incorrect, misleading and inconsistent with the data in other experiments (Fig. 6e and the new Supplementary Figure 6a). In response to this concern, the authors now claim they have repeated the experiment. Inexplicably, now they don’t see any expression of Dnase2 or Trex1. Yet again they go on to state “HDFs depleted of cGAS or STING by siRNA failed to increase the expression levels of 250 DNase2 and TREX1 (Fig. 2a)”. Again, this is misleading since what they are referring to is a comparison of no signal vs no signal. That notwithstanding, as already mentioned these particular data are not consistent with the data in the other experiments (Fig. 6e and the new Supplementary Figure 6a). The authors’ response is that they don’t know. Then they argue that this could be due to the limitations of shRNA silencing compared to knockout cells. Which of their shRNA experiments should the reader therefore believe? If they themselves have doubts about some components of the data, then why are they included?

Response:

We believe that we can discuss several possibilities for the explanation of our results in the discussion part, and we described one possibility. So, we do not understand what the problem is in our discussion. We would like to stress that the RNAi treatment is a knockdown, but not a knockout. So, it is important to bear in mind that the residual levels of target gene expression may still have some functions in the RNAi experiments. Nevertheless, RNAi technology is a useful tool to analyze gene function and is still widely used in many biological studies.

And by the way, why did they not include the corresponding data in Fig. 2b - these are obvious controls? Back to Fig. 2a: although not optimal, the previous DNase and Trex1 plots in Fig. 2a at least showed some signal. So, on what basis have the authors now decided that these new blots without any signal are the correct ones to include and not the previous ones? This seems like an attempt to cherry pick and present data to detract the reader from the apparent inter-experimental consistencies. This casts serious doubts about the reproducibility of their experiments. They claim that their data are representative of three independent experiments. If that were the case, upon request they should have readily provided an entire new set of data from an independent biological replicate. Instead, they have done the very minimum; present reanalysis of samples from one experiment, which even then turns out to be inconclusive.

I can highlight many other examples but I don't believe this is the point of the review process. My part, I believe, is to point out the main issues the authors need to pay attention to and hope they address them fully, and not just the very minimum to go past the reviewer. This does not improve the manuscript and I feel this is the case!

Response:

We are very sorry for the confusion. Since the DNase2 and Trex1 (lanes 2-4 in the original version of Fig. 2a) signals are very faint and hardly detectable as compared to the control (lane 1), we thought that there might have been some contamination during the sample loading step for western blotting. We therefore repeated the western blotting for DNase2/Trex1, using exactly the same samples, and found that the faint signals of DNase2/Trex1 that we observed in the original Fig. 2a are likely due to some contamination. However, of course we also repeated the same experiment using a completely different set of the samples. An example is shown below.

Previous Comment: “In contrast to their observations, a number studies have previously shown that Trex1 is in fact upregulated in response to DNA damage (e.g. Vanpouille-Box C et al., *Nat Commun.* 2017;8:15618, Tomicic MT et al, *Biochim Biophys Acta.* 2013;1833(8), Christmann M et al., *Nucleic Acids Res.* 2010;38(19)) and that such induction of Trex1 occurs via cytokine-induced transcription factors such as STAT1 and Fos/AP-1 (Christmann M et al. *Nucleic Acids Res.* 2010;38(19), Serra M et al. *J Immunol.* 2011; 186(4)). Interestingly, cytokine-induced senescence has also recently been linked to the activation of p16-RB and STAT1 pathways (Braumüller H et al., *Nature.* 2013, 494(7437)). These published findings do not necessarily discount the proposed downregulation of Trex1 and DNase2 in senescent cells. Rather they illustrate that both DNA damage and cytokine signals can regulate the expression of cytoplasmic DNases. Whether upregulation or downregulation most likely depend on the context. In the present study, the authors analysed the expression of cytoplasmic DNases in serially passaged or oncogene-induced senescent cells and conclude that DNA damage induces the downregulation DNase2 and Trex1. What is the immediate impact of DNA damage (e.g. 1-6 hours after exposure DNA damaging agents) on the expression these DNases? This could have been a good control to include. Therefore, given the experimental conditions of their tests, it is not possible to conclude whether the proposed regulation of DNases is directly due to DNA damage per se or an outcome of the SASP. As aforementioned, the data in Figure 2d (now Fig. 4a) and Figure 4e (now Fig. 6e) are indicative of the latter. In brief, the central message of the manuscript “DNA damage response provokes SASP by downregulating cytoplasmic DNases in senescent cells” is an overstatement”.

In response to this comment the authors have provided in the rebuttal an experiment where

they show the expression of DNase 2, Trex1 and other activation markers during a 2-10 days course of Ras-induced senescence. This does not address the issue. The expression changes they are showed are most likely due to secondary effects and not direct consequences of DNA damage which is precisely what I requested for: 1-6 hours after exposure of cells to DNA damaging agents such as ionizing irradiation.

Comment: I thank the authors for finally performing this control experiment which should be included at least as part of the supplementary figures. Although the authors claim that there is some decrease, clearly there is no decreased expression of Dnase 2 or Trex1 upon DNA damage until after 10 days. This confirms that activation of p16-RB pathway or the downregulation of Dnase 2 and Trex1 is NOT a direct consequence of DNA damage but a secondary outcome of senescence. The authors argue that decrease in Dnase2 and Trex1 is not apparent in cells because “It takes at least 6 days to detect cellular senescence after oncogenic Ras-expression, as judged by p16 expression in cultured HDFs”. I take this as an acknowledgement of the same: activation of p16-RB pathway is not directly due to DNA damage but a secondary consequence of senescence.

Response:

Thank you for your understanding. As we had mentioned, cellular senescence occurs in the late stage of the DNA damage response, and not in the immediate early stage of the DNA damage response.